# LEARNING WITHOUT FORGETTING: TASK AWARE MULTI-TASK LEARNING FOR MULTI-MODALITY TASKS

## ABSTRACT

Existing Multi-Task Learning(MTL) strategies like joint or meta-learning focus more on shared learning and have little to no scope for task-specific learning. This creates the need for a distinct shared pretraining phase and a task-specific finetuning phase. The finetuning phase creates separate models for each task, where improving the performance of a particular task necessitates forgetting some of the knowledge garnered in other tasks. Humans, on the other hand, perform task-specific learning in synergy with general domain-based learning. Inspired by these learning patterns in humans, we suggest a simple yet generic task aware framework to incorporate into existing MTL strategies. The proposed framework computes task-specific representations to modulate the model parameters during MTL. Hence, it performs both shared and task-specific learning in a single phase resulting in a single model for all the tasks. The single model itself achieves significant performance gains over the existing MTL strategies. For example, we train a model on Speech Translation (ST), Automatic Speech Recognition (ASR), and Machine Translation (MT) tasks using the proposed task aware multitask learning approach. This single model achieves a performance of 28.64 BLEU score on ST MuST-C English-German, WER of 11.61 on ASR TEDLium v3, and BLEU score of 23.35 on MT WMT14 English-German tasks. This sets a new state-of-the-art performance (SOTA) on the ST task while outperforming the existing end-to-end ASR systems with a competitive performance on the MT task.

## 1 INTRODUCTION

The process of Multi-Task Learning (MTL) on a set of related tasks is inspired by the patterns displayed by human learning. It involves a pretraining phase over all the tasks, followed by a finetuning phase. During pretraining, the model tries to grasp the shared knowledge of all the tasks involved, while in the finetuning phase, task-specific learning is performed to improve the performance. However, as a result of the finetuning phase, the model forgets the information about the other tasks that it learnt during pretraining. Humans, on the other hand, are less susceptible to forgetfulness and retain existing knowledge/skills while mastering a new task. For example, a polyglot who masters a new language learns to translate from this language without losing the ability to translate other languages. Moreover, the lack of task-based flexibility and having different finetuning/pretraining phases cause gaps in the learning process due to the following reasons:

**Role Mismatch:** Consider the MTL system being trained to perform the Speech Translation(ST), Automatic Speech Recognition(ASR) and Machine Translation(MT) tasks. The Encoder block has a very different role in the standalone ASR, MT and ST models and hence we cannot expect a single encoder to perform well on all the tasks without any cues to identify/use task information. Moreover, there is a discrepancy between pretraining and finetuning hampering the MTL objective.

**Task Awareness:** At each step in the MTL, the model tries to optimize over the task at hand. For tasks like ST and ASR with the same source language, it is impossible for the model to identify the task and alter its parameters accordingly, hence necessitating a finetuning phase. A few such examples have been provided in Table 1. Humans, on the other hand, grasp the task they have to perform by means of context or explicit cues.

| Task | Confounding Task | MTL Prediction | Task Aware Output |
|:---:|:---:|:---:|:---:|
| En ASR | En-De ST | sogar ihre eigenen Eltern Mobilität opportunity to.. | even their own parents Mobility opportunity to .. |
| En-De MT | En-Ro MT | Situația este alarmantă | Die Situation ist alarmierend |
| En-De ST | En ASR | I don't understand it. was a success or not. | Das verstehe ich nicht war ein Erfolg oder nicht. |

Table 1: Issues due to the lack of task information in MTL strategies. The MTL model, unable to identify the task, produces the output corresponding to another task either completely or partially.(En:English, De:German, Ro:Romanian). Task aware output is the output obtained from our proposed approach.

Although MTL strategies help the finetuned models to perform better than the models directly trained on those tasks, their applicability is limited to finding a good initialization point for the finetuning phase. Moreover, having a separate model for each task increases the memory requirements, which is detrimental in low resource settings.

In order to achieve the goal of jointly learning all the tasks, similar to humans, we need to perform shared learning in synergy with task-specific learning. Previous approaches such as Raffel et al. (2019) trained a joint model for a set of related text-to-text tasks by providing the task information along with the inputs during the joint learning phase. However, providing explicit task information is not always desirable, e.g., consider the automatic multilingual speech translation task. In order to ensure seamless user experience, it is expected that the model extracts the task information implicitly.

Thus, a holistic joint learning strategy requires a generic framework which learns task-specific information without any explicit supervision.

In this work, we propose a generic framework which can be easily integrated into the MTL strategies which can extract task-based characteristics. The proposed approach helps align existing MTL approaches with human learning processes by incorporating task information into the learning process and getting rid of the issues related to forgetfulness. We design a modulation network for learning the task characteristics and modulating the parameters of the model during MTL. As discussed above, the task information may or may not be explicitly available during the training. Hence, we propose two different designs of task modulation network to learn the task characteristics; one uses explicit task identities while the other uses the examples from the task as input. The model, coupled with the modulation network, jointly learns on all the tasks and at the same time, performs the task-specific learning. The proposed approach tackles issues related to forgetfulness by keeping a single model for all the tasks, and hence avoiding the expensive finetuning phase. Having a single model for all the tasks also reduces memory constraints, improving suitability for low resource devices.

To evaluate the proposed framework, we conduct two sets of experiments. First, we include the task information during MTL on text-to-text tasks to show the effect of task information. Secondly, we train a model on tasks with different modalities and end goals, with highly confounding tasks. Our proposed framework allows the model to learn the task characteristics without any explicit supervision, and hence train a single model which performs well on all the tasks. The main contributions of this work are as follows:

- We propose an approach to tackle the issue of forgetfulness which occurs during the finetuning phase of existing MTL strategies.

- Our model, without any finetuning, achieves superior performance on all the tasks which alleviates the need to keep separate task-specific models.

- Our proposed framework is generic enough to be used with any MTL strategy involving tasks with multiple modalities.

## 2 TASK-AWARE MULTITASK LEARNING

An overview of our proposed approach is shown in Figure 1.

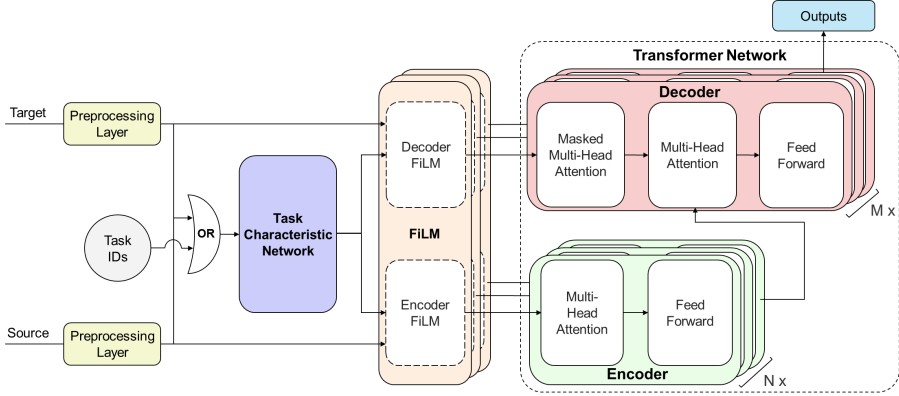

Figure 1: Overview of task aware multi-task learning. Task Characteristics Network (TCN) produces a task embedding to modulate the parameters of base network. Inputs to the TCN are chosen based on type of TCN.

## 2.1 BASE MODEL

In general, the sequence-to-sequence architecture consists of two components: (1) an encoder which computes a set of representations $X = \{x_1, \cdots, x_m\} \in \mathbb{R}^{m \times d}$ corresponding to $x$, and a decoder coupled with attention mechanism (Bahdanau et al., 2015) dynamically reads encoder's output and predicts target language sequence $Y = \{y_1, \cdots, y_n\} \in \mathbb{R}^{n \times d}$. It is trained on a dataset $D$ to maximize the $p(Y|X; \theta)$, where $\theta$ are parameters of the model.

We use the Transformer Vaswani et al. (2017) as our base model. Based on the task modalities, we choose the preprocessing layer in the Transformer, i.e., speech or the text (text-embedding) preprocessing layer. The speech preprocessing layer consists of a stack of $k$ CNN layers with stride 2 for both time and frequency dimensions. This layer compresses the speech sequence and produces the output sequence such that input sequences corresponding to all the tasks have similar dimensions, $d$. The overview of the base sequence-to-sequence model is shown in the rightmost part of Figure 1.

## 2.2 TASK MODULATION NETWORK

The task modulation network performs two operations. In the first step, it computes the task characteristics ($t_e$) using the task characteristics layer. It then modulates the model parameters $\theta$ using $t_e$ in the second step.

### 2.2.1 TASK CHARACTERISTICS NETWORK:

We propose two types of Task Characteristics Networks(TCN) to learn the task characteristics, where one uses explicit task identities while the other uses source-target sequences as input.

**Explicit Task Information:** In this approach, the tasks involved are represented using different task identities and fed as input to this TCN as one hot vectors. This network consists of a feed-forward layer which produces the task embedding used for modulating the model parameters.

$$t_e = FFN(e), \tag{1}$$

where $e \in \mathbb{R}^s$ is a one-hot encoding of $s$ tasks used during joint learning.

**Implicit Task Information:** The Implicit TCN computes the task embeddings using example sequences from the tasks without any external supervision. It consists of four sub-layers: (1) Sequence Representation Layer, (2) Bi-directional Attention Layer, (3) Sequence Summary Layer, and (4) Task Embedding Layer.

The sequence representation sub-layer consists of uni-directional Transformer Encoder (TE) blocks Vaswani et al. (2017). It takes the source and target sequences from the tasks as input and produces

self-attended source and target sequences.

$$\boldsymbol{X}^{sa} = \text{TE}(\boldsymbol{X}), \quad \boldsymbol{Y}^{sa} = \text{TE}(\boldsymbol{Y}), \tag{2}$$

where $\boldsymbol{X}^{sa} \in \mathbb{R}^{M \times d}$, $\boldsymbol{Y}^{sa} \in \mathbb{R}^{N \times d}$. This sub-layer computes the contextual representation of the sequences.

The Bi-directional Attention (BiA) sub-layer takes the self-attended source and target sequences from the previous layer as input and computes the relation between them using Dot-Product Attention Luong et al. (2015). As a result, we get target aware source ($\boldsymbol{X}^{at} \in \mathbb{R}^{M \times d}$) and source aware target ($\boldsymbol{Y}^{as}\mathbb{R}^{N \times d}$) representations as outputs.

$$\boldsymbol{X}^{at} = \text{BiA}(\boldsymbol{X}^{sa}, \boldsymbol{Y}^{sa}), \quad \boldsymbol{Y}^{as} = \text{BiA}(\boldsymbol{Y}^{sa}, \boldsymbol{X}^{sa}). \tag{3}$$

The sequence summary sub-layer is similar to the sequence representation sub layer and summarizes the sequences. The sequence summaries are given by:

$$\boldsymbol{X}^{s} = \text{TE}_{\text{u}}(\boldsymbol{X}^{at}), \quad \boldsymbol{Y}^{s} = \text{TE}_{\text{u}}(\boldsymbol{Y}^{as}), \tag{4}$$

where $\boldsymbol{X}^{s} \in \mathbb{R}^{M \times d}$, $\boldsymbol{Y}^{s} \in \mathbb{R}^{N \times d}$. The Equation 4 summarizes the sequences $\boldsymbol{X}^{at}$ and $\boldsymbol{Y}^{as}$ which contain the contextual and attention information. We take the last tokens from both the $\boldsymbol{x}^{s} \in \mathbb{R}^{d}$ and $\boldsymbol{y}^{s} \in \mathbb{R}^{d}$, since the last token can see the whole sequence and acts as a summary of the sequence.

The task embedding layer computes $\boldsymbol{t_e}$ by taking the outputs of the sequence summary sub-layer and applying a feed-forward network:

$$\boldsymbol{t_e} = FFN([\boldsymbol{x}^{s} : \boldsymbol{y}^{s}]). \tag{5}$$

### 2.2.2 MODULATING MODEL PARAMETERS

We modulate the parameters ($\theta$) of the network (Section 2.1) to account for the task-specific variation during MTL over a set of tasks. We achieve this by scaling ($\boldsymbol{\gamma}$) and shifting ($\boldsymbol{\beta}$) the outputs of each layer (e.g., transformer block) including any preprocessing layers in the model adopted based on the Feature-wise Linear Modulation (FiLM; Perez et al. (2018)). The $\boldsymbol{\gamma}$ and $\boldsymbol{\beta}$ parameters are obtained from the task embedding $\boldsymbol{t_e}$ either by using equation 1 or 5.

$$\boldsymbol{\gamma} = \boldsymbol{t_e}[: d], \quad \boldsymbol{\beta} = \boldsymbol{t_e}[d :], \tag{6}$$

where $\boldsymbol{t_e} \in \mathbb{R}^{2d}$, and $d$ is the hidden dimension of the model.

Once we have $\boldsymbol{\gamma}$ and $\boldsymbol{\beta}$, we apply the feature-wise linear modulation (Perez et al., 2018) to compute the modulated output ($O_l$) for each block of the model.

$$O_l = \boldsymbol{\gamma} * f_l(v_l; \theta_l) + \boldsymbol{\beta}, \quad l = 1, \cdots, L, \tag{7}$$

where $L$ is the total number of blocks in the model and $f_l$ represents the $l_{th}$ block of the model with parameters $\theta_l \in \theta$ and inputs $v_l$.

### 2.3 TRAINING

MTL has been successfully applied across different applications of machine learning such as natural language processing (Hashimoto et al., 2016; Collobert & Weston, 2008), speech recognition (Liu et al., 2019; Deng et al., 2013), computer vision (Zhang et al., 2014; Liu et al., 2015; Girshick, 2015), and drug discovery (Ramsundar et al., 2015). It comes in many forms: joint learning, learning to learn, and learning with auxiliary tasks. We consider two MTL strategies: (1) joint learning and (2) learning to learn to train on set of $S$ tasks, $\text{T} = \{\tau^1, \cdots, \tau^S\}$ with corresponding datasets $D = \{D^1, \cdots, D^S\}$.

As our first training strategy, we use Joint Learning (JL) (Caruana, 1997), which is the most commonly used training strategy for MTL. In JL, the model parameters, including the output layer, are shared across all the tasks involved in the training. For the second training strategy under the learning-to-learn approach, we use a variant of meta-learning, Modality Agnostic Meta Learning (MAML) (Finn et al., 2017a). Even though MAML is mostly used in few-shot learning settings, we use it since it

| S. No. | MTL Strategy | Training Approach | En-De (↑) | En-Tr((↑) | En-Ro((↑) |
|---|---|---|---|---|---|
| 1 | Joint Learning | Vanilla | 7.65 | 3.59 | 5.55 |
| 2 | | OHV(Raffel et al., 2019)* | 16.77 | 6.10 | 11.79 |
| 3 | Meta Learning | MAML (Finn et al., 2017a)* | 7.74 | 2.93 | 5.78 |
| 4 | | OHV(Raffel et al., 2019)* | 16.08 | 5.93 | 12.43 |
| **This Work** | | | | | |
| 5 | Joint Learning | Implicit TCN | 14.18 | 4.38 | 13.47 |
| 6 | Meta Learning | Implicit TCN | 14.50 | 4.43 | 14.32 |

Table 2: Performance(BLEU) of the models trained on the Single Modality for the En-De, En-Ro and En-Tr Machine Translation task. *Models based on OHV and MAML are inspired from Raffel et al. (2019) and Finn et al. (2017a).

allows for task-specific learning during the meta-train step and it has also been shown to provide improvements in the field of speech translation(Indurthi et al., 2020).

We resolve the source-target vocabulary mismatch across different tasks in MTL by using a vocabulary of subwords (Sennrich et al., 2016) computed from all the tasks. We sample a batch of examples from $D^s$ and use this as input to the TCN and the Transformer model. To ensure that each training example uses the task embedding computed using another example, we randomly shuffle this batch while using them as input to the TCN. This random shuffling improves the generalization performance by forcing the network to learn task-specific characteristics ($t_e$) in Equation 1 or 5. We compute the task embedding in the meta-train step as well; however, the parameters of the TCN are updated only during the meta-test step. During inference time, we use the precomputed task embeddings using a batch of examples randomly sampled from the training set.

## 3 EXPERIMENTS

### 3.1 TASKS AND DATASETS

We conduct two sets of experiments, one with the tasks having the same input modality, i.e., text and another over tasks having different input modalities, i.e., speech and text. The main motivation behind the text-based experiments is to establish the importance of providing task information in MTL. Our main experiments, containing different input modalities involve highly confusing tasks. These experiments help us demonstrate the effectiveness of our approach in a generic setup. We incorporate the proposed task modulation framework into joint and meta-learning strategies and analyze its effects.

#### 3.1.1 SINGLE MODALITY EXPERIMENTS

We perform the small scale text-to-text machine translation task over three language pairs English-German/Romanian/Turkish (En-De/Ro/Tr). We keep English as the source language, which makes it crucial to use task information and produce different outputs from the same input. Since it is easier to provide task identity through one-hot vectors in text, we provide the task information by simply prepending the task identity to the source sequence of each task, e.g., "*translate from English to German*", "*translate from English to Turkish*" similar to Raffel et al. (2019). We also train models using our proposed framework to learn the task information and shared knowledge jointly.

For En-De, we use 1.9M training examples from the `Europarl v7` dataset. `Europarl dev2006` and `News Commentary nc-dev2007` are used as the dev and `Europarl devtest2006`, `Europarl test2006` and `News Commentary nc-devtest2007` as the test sets. For En-Tr we train using 200k training examples from the `setimes2` dataset. We use `newsdev2016` as the dev and `newstest2017` as the test set. For En-Ro, we use 600k training examples from `Europarl v8` and `setimes2` datasets. We use `newsdev2016` as dev and `newstest2016` as the test set.

| S. No. | MTL Strategy | Training | ST (↑) | ASR(↓) | MT (↑) |
|--------|--------------|----------|--------|--------|--------|
| 1 | N/A | Direct Learning | 22.14 | 25.25 | 27.93 |
| 2 | Joint Learning | Pretraining | 0.81 | 14.83 | 23.62 |
| 3 | | + Finetuning | 25.99 | 12.95 | 22.63 |
| 4 | Meta Learning | Pretraining | 0.51 | 32.93 | 11.40 |
| 5 | (Indurthi et al., 2020) | + Finetuning | 26.03 | 12.83 | 22.15 |
| **This Work** | | | | | |
| 6 | Joint Learning | Explicit TCN | 28.34 | 11.99 | 23.15 |
| 7 | | Implicit TCN | **28.64** | **11.61** | 23.35 |
| 8 | | Taskwise Best Models | 28.88 | 11.36 | 23.71 |
| 9 | Meta Learning | Explicit TCN | 28.16 | 11.68 | 23.13 |
| 10 | | Implicit TCN | **28.55** | **11.65** | 23.41 |
| 11 | | Taskwise Best Models | 28.59 | 11.28 | 23.57 |

Table 3: Performance of the models trained on the Multiple Modality tasks, i.e., ST (MuST-C En-De), ASR (TED-LIUM 3), and MT (WMT'15 En-De).

### 3.1.2 Multiple Modality Experiments

To alleviate the data scarcity issue in Speech Translation (ST), several MTL strategies have been proposed to jointly train the ST task with Automatic Speech Recognition (ASR) and Machine Translation (MT) tasks. These MTL approaches lead to significant performance gains on both ST and ASR tasks after the finetuning phase. We evaluate our proposed framework based on this multimodal MTL setting since passing the task information explicitly via prepending labels(like the text-to-text case) in the source sequence is not possible. We use the following datasets for ST English-German, ASR English, MT English-German tasks:

**MT En-De:** We use the Open Subtitles (Lison et al., 2019) and WMT 19 corpora. WMT 19 consists of Common Crawl, Europarl v9, and News Commentary v14 datasets(22M training examples).

**ASR English**: We used five different datasets namely LibriSpeech (Panayotov et al., 2015), MuST-C (Di Gangi et al., 2019), TED-LIUM (Hernandez et al., 2018), Common Voice (Ardila et al., 2020) and filtered IWSLT 19 (IWS, 2019) to train the English ASR task.

**ST Task:** We use the Europarl ST (Iranzo-Sánchez et al., 2019), IWSLT 2019 (IWS, 2019) and MuST-C (Di Gangi et al., 2019) datasets. Since ST task has lesser training examples, we use data augmentation techniques (Lakumarapu et al., 2020) to increase the number of training examples.

Please refer to the appendix for more details about the data statistics and data augmentation techniques used. All the models reported in this work use the same data settings for training and evaluation.

### 3.2 Implementation Details and Metrics

We implemented all the models using Tensorflow 2.2 framework. For all our experiments, we use the Transformer(Vaswani et al., 2017) as our base model. The hyperparameter settings such as learning rate, scheduler, optimization algorithm, and dropout have been kept similar to the Transformer, other than the ones explicitly stated to be different. The ASR performance is measured using Word Error Rate (WER) while ST and MT performances are calculated using the detokenized cased BLEU score (Post, 2018). We generate word-piece based universal vocabulary (Gu et al., 2018a) of size 32k using source and target text sequences of all the tasks. For the task aware MTL strategies, we choose a single model to report the results rather than finding the best model for each task separately.

We train the text-to-text translation models using 6 Encoder and Decoder layers with a batch size of 2048 text tokens. The training is performed using NVIDIA P40 GPU for 400k steps.

In multi-modality experiments, the speech signals are represented using 80-dimensional log-Mel features and use 3 CNN layers in the preprocessing layer described in Section 2.1. We use 12 Encoder and Decoder layers and train for 600k steps using 8 NVIDIA V100 GPUs. For the systems without TCN, we perform finetuning for 10k steps on each task.

| S. No. | Task | Approach | (BLEU(↑)/WER(↓)) |
|---|---|---|---|
| 1 | MT (WMT'15 En-De) | Liu et al. (2020)(60 Encoders) | 30.10 |
| 2 | | Our Approach(12 Encoders) | 23.71 |
| 3 | ST (MuSTC En-De) | Indurthi et al. (2020) | 22.11 |
| 4 | | Pino et al. (2020) | 25.99 |
| 5 | | Lakumarapu et al. (2020) | 27.51 |
| 6 | | Our Approach | **28.88** |
| 7 | ASR (TED-LIUM 3) | Pham et al. (2019)(36 Encoders) | 10.20 |
| 8 | | Pham et al. (2019)(12 Encoders) | 12.40 |
| 9 | | Our Approach(12 Encoders) | **10.01** |

Table 4: Performance comparison with the existing works in ASR, ST and MT.

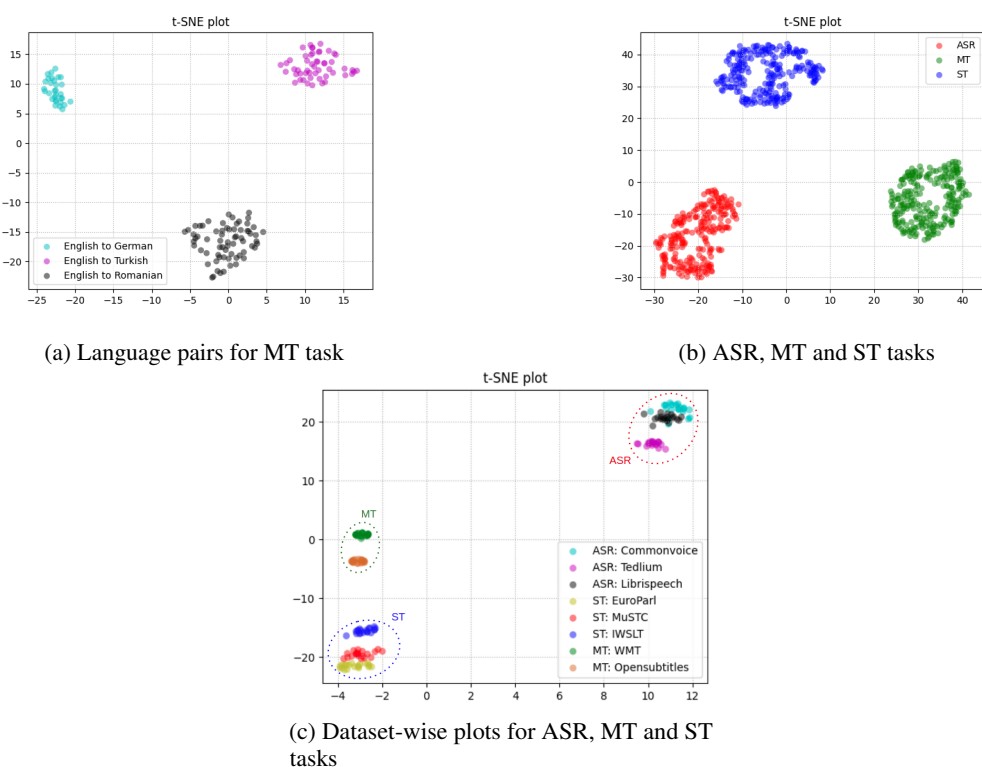

(a) Language pairs for MT task

(b) ASR, MT and ST tasks

(c) Dataset-wise plots for ASR, MT and ST tasks

Figure 2: t-SNE plot of the task modulation output shows a clear demarcation between various tasks and datasets.

## 3.3 RESULTS

### 3.3.1 SINGLE MODALITY EXPERIMENTS

The results for the text-to-text translation models trained with different MTL strategies have been provided in Table 2. The MTL models with prepended task label (Raffel et al., 2019) are referred to as OHV (One Hot Vector). Unlike T5, we don't initialize the models with the text embeddings from large pretrained language model (Devlin et al., 2018). Instead, we focus on establishing the importance of task information during MTL and having a single model for all the tasks. As we can see from the results, providing the task information via text labels or implicitly using the proposed task aware MTL leads to significant performance improvements compared to the MTL without the task information. The models trained using OHV have better performance than those trained using implicit TCN. However, providing OHV via text labels is not always possible for tasks involving non-text modalities such as speech and images.

### 3.3.2 Multi Modality Experiments

We evaluate the proposed two TCNs and compare them with the vanilla MTL strategies. The performance of all the models is reported in Table 3. We also extended the T5 (Raffel et al., 2019) approach to the multi modality experiments and compare it with our approach.

**Effect of Task Information:** The models trained using task aware MTL achieve significant performance gains over the models trained using vanilla MTL approach. Our single model achieves superior performance compared to the vanilla MTL models even after the finetuning. This shows that not only is the task information essential to identify the task, but also helps to extract the shared knowledge better. Our JL and MAML models trained with task aware MTL achieve improvements of (+2.65, +2.52) for ST, (-1.34, -1.18) for ASR, and (+0.72, +1.26) for the MT task. MAML has some scope for task-specific learning during its meta train step, which explains why the improvements for MAML are slightly lesser than JL for ST and ASR tasks.

We also report results using Direct Learning (DL) approach, where separate models are trained for each task, to compare with MTL models. All the MTL models outperform the DL models on ST and ASR tasks have comparable performance on MT task.

**Explicit v/s Implicit TCN:** Our proposed implicit TCN learns the task characteristics directly from the examples of each task and achieves a performance comparable to the models trained using explicit TCN. This indicates that it is better to learn the task information implicitly, specifically for tasks having overlapping characteristics. Figure 2 contains the tSNE plots for task embeddings obtained from the implicit TCN for single and multi-modality experiments. We can observe that the implicit TCN is also able to separate all the three tasks effectively without any external supervision.

**Single model for all tasks:** We select one single model for reporting the results for our approach, since, having a single model for multiple tasks is favourable in low resource settings. However, we also report the best models corresponding to each task (row 8 and 11 of Table 3). We observe that choosing a single model over task-specific models did not result in any significant performance loss.

**Feature-wise v/s Input based modulation:** We also implemented the input based conditioning (Toshniwal et al., 2018; Raffel et al., 2019) where we prepend the TCN output, i.e., task information to the source and target sequences. As compared to our approach, this approach provides a comparable performance on the ASR task. However, the ST performance is erratic and the output is mixed between ST and ASR tasks. This shows that the feature-wise modulation is more efficient way to carry out task-based conditioning for highly confusing tasks like ST and ASR.

**Number of parameters added:** The Explicit TCN, which is a dense layer, roughly 1500 new parameters are added. For the Implicit TCN, roughly 1 million new additional parameters are added. However, simply increasing the number of parameters is not sufficient to to improve the performance. For e.g., we trained several models by increasing the number of layers for encoder and decoder upto 16. However, these models gave inferior performance as compared to the reported models with 12 encoder and decoder layers.

**Scaling with large number of tasks:** The t-sne plots in the Figure 2b are drawn using the three test datasets. However, we used multiple datasets for each of the ASR(Librispeech, Common voice, TEDLIUM, MuSTC-ASR), ST (MuSTC, IWSLT20, Europarl), and MT (WMT19, OpenSubtitles) tasks in the multi-modality experiments. We analyze whether or not our proposed approach is able to separate the data coming from these different distributions. As compared to data coming from different tasks, separating the data coming from the same task(generated from different distributions) is more difficult. Earlier, in Figure 2b, we observed that the output is clustered based on the tasks. Figure 2c shows that within these task-based clusters, there are sub-clusters based on the source dataset. Hence, the model is able to identify each sub-task based on the source dataset. The model also gives decent performances on all of them. For example, the single model achieves a WER of 7.5 on the Librispeech tst-clean, 10.35 on MuSTC, 11.65 on the TEDLIUM v3 and 20.36 on the commonvoice test set. For the ST task, the same model gives a BLEU score of 28.64 on the MuSTC test set, 27.61 on the IWSLT tst-2010, and 27.57 on the Europarl test set. This shows that our proposed approach scales well with the total number of tasks.

**Comparison with existing works:** The design of our system, i.e., the parameters and the related tasks were fixed keeping the ST task in mind. We compare the results of our best systems(after checkpoint averaging) with the recent works in Table 4. We set a new state-of-the-art (SOTA) on the

ST En-De MuST-C task. For the ASR task, we outperform the very deep Transformer based model Pham et al. (2019). We achieve a 19.2% improvement in the WER as compared to the model with the same number of Encoder and Decoder blocks. The best transformer-based MT model achieves a BLEU score of 30.10, however, it uses 60 Encoder blocks. The performance drop on the MT task is attributed to simply training a bigger model without using any additional initialization techniques proposed in Liu et al. (2015); Wu et al. (2019). However, the MT task helps the other tasks and improves the overall performance of the system.

## 4 RELATED WORK

Various MTL techniques have been widely used to improve the performance of end-to-end neural networks. These techniques are known to solve issues like overfitting and data scarcity. Joint learning (Caruana, 1997) improves the generalization by leveraging the shared information contained in the training signals of related tasks. MAML (Finn et al., 2017b) was proposed for training a joint model on a variety of tasks, such that it can quickly adapt to new tasks. Both the learning approaches require a finetuning phase resulting in different models for each task. Moreover, during finetuning phase the model substantially forgets the knowledge acquired during the large-scale pretraining.

One of the original solutions to this problem is pseudo-rehearsal, which involves learning the new task while rehearsing generated items representative of the previous task. This has been investigated and addressed to a certain extent in Atkinson et al. (2018) and Li & Hoiem (2018). He et al. (2020) address this by using a mix-review finetuning strategy, where they include the pretraining objective during the finetuning phase. Raffel et al. (2019) take a different approach by providing the task information to the model and achieve performance improvements on different text-to-text tasks. Although this alleviates the need for finetuning, it cannot be extended to the tasks involving complex modalities. In our work, we propose a generic framework on top of MTL to provide task information to the model which can be applied irrespective of the task modalities. It also removes the need for finetuning, tackling the issue of forgetfulness at its root cause.

A few approaches have also tried to train multiple tasks with a single model, Cheung et al. (2019) project the input to orthogonal sub-spaces based on the task information. In the approach proposed by Li & Hoiem (2018), the model is trained on various image classification tasks having the same input modality. They preserve the output of the model on the training example such that the parameters don't deviate much from the original tasks. This is useful when the tasks share the same goal, e.g. classification. However, we train on a much more varied set of tasks, which might also have the same inputs with different end goals. Strezoski et al. (2019) propose to apply a fixed mask based on the task identity. Our work can be seen as a generalization of this work. As compared to all these approaches, our model is capable of performing both task identification and the corresponding task learning simultaneously. It learns to control the interactions among various tasks based on the inter-task similarity without any explicit supervision.

In the domain of neural machine translation, several MTL approaches have been proposed (Gu et al., 2018a;b). Similarly, recent works have shown that jointly training ST, ASR, and MT tasks improved the overall performance (Liu et al., 2019; Indurthi et al., 2020). However, all these require a separate finetuning phase.

## 5 CONCLUSION

This work proposes a task-aware framework which helps to improve the learning ability of the existing multitask learning strategies. It addresses the issues faced during vanilla multitask learning, which includes forgetfulness during finetuning and the problems associated with having separate models for each task. The proposed approach helps to align better the existing multitask learning strategies with human learning. It achieves significant performance improvements with a single model on a variety of tasks which is favourable in low resource settings.

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

## 6    APPENDIX

### 6.1    DATASETS

#### 6.1.1    DATA AUGMENTATION FOR SPEECH TRANSLATION

Table 5 provides details about the datasets used for the multi-modality experiments. Since En-De ST task has relatively fewer training examples compared to ASR and MT tasks, we augment the ST dataset with synthetic training examples. We generate the synthetic speech sequence and pair it with the synthetic German text sequences. obtained by using the top two beam search results of the two trained English-to-German NMT models. For speech sequence, we use the Sox library to generate the speech signal using different values of speed, echo, and tempo parameters similar to (Potapczyk et al., 2019). The parameter values are uniformly sampled using these ranges : tempo $\in (0.85, 1.3)$, speed $\in (0.95, 1.05)$, echo_delay $\in (20, 200)$, and echo_decay $\in (0.05, 0.2)$. We also train two NMT models on EN-De language pair to generate synthetic German sequence. The first model is based on Edunov et al. (2018) and the second model (Indurthi et al., 2019) is trained on WMT'18 En-De and *OpenSubtitles* datsets. We increase the size of the IWSLT 19(filtered) ST dataset to five times of the original size by augmenting 4x data – four text sequences using the top two beam results from each EN-De NMT model and four speech signals using the Sox parameter ranges. For the Europarl-ST, we augment 2x examples to triple the size. The TED-LIUM 3 dataset does not contain speech-to-text translation examples originally; hence, we create 2x synthetic speech-to-text translations using speech-to-text transcripts. Finally, for the MuST-C dataset, we only create synthetic speech and pair it with the original translation to increase the dataset size to 4x. The Overall, we created the synthetic training data of size approximately equal to four times of the original data for the ST task.

#### 6.1.2    TASK IDENTIFICATION WITHOUT TASK INFORMATION

Under the multi-modality setting, we conducted smaller scale experiments using only one dataset for each ST, ASR, and ST tasks. The details of the datasets used have been provided in Table 7. We trained on single p40 GPU for 400k steps. The corresponding results have been reported in Table 6. All the results have been obtained without any finetuning. Even though our task-aware MTL model achieves significant performance improvement over vanilla MTL models, we can observe that the vanilla MTL models are also able to give a decent performance on all tasks without any finetuning. An explanation for this is that we used MuST-C dataset for the En-De ST task and TEDLium v3 for the ASR task, which means that the source speech is coming from 2 different sources. However, if we use the same datasets for both the tasks(after data augmentation), the MTL models get confused and the ST, ASR outputs are mixed. The MTL models might be able to learn the task identities simply based on the source speech sequences, since these sequence are coming from different datasets for each task type–MuST-C for ST and TED-LIUM v3 for ASR. However, this does not mean that vanilla MTL models perform joint learning effectively. A human who can perform multiple tasks from the same input is aware of the task he has to perform beforehand. Similarly, it is unreasonable to expect different outputs (translation, transcription) from a model to the same type of input (English speech) without any explicit task information.

#### 6.1.3    IMPLEMENTATION DETAILS

The detailed hyperparameters settings used for the single modality and multi modality experiments have been provided in the Table 8.

| Task | Corpus | # hours | # Examples |
|---|---|---|---|
| MT | Open Subtitles | N/A | 22,512,639 |
| MT | WMT 19 | N/A | 4,592,289 |
| ASR | LibriSpeech | 982 | 232,958 |
| ASR | IWSLT 19 ST(filtered) | 220 | 145,372 |
| ASR | MuST-C | 400 | 229,702 |
| ASR | CommonVoice | 1469 | 232,958 |
| ASR | TED-LIUM 3 | 452 | 286,263 |
| ST | Europarl-ST | 89 | 32,628 |
| ST | IWSLT 19 ST(filtered) | 220 | 145,372 |
| ST | MuST-C | 400 | 229,703 |

Table 5: Number of original training examples in each dataset.

| S No. | MTL Strategy | MT BLEU ($\uparrow$) Test | ASR(WER ($\downarrow$)) Dev | ASR(WER ($\downarrow$)) Test | ST(BLEU ($\uparrow$)) Dev | ST(BLEU ($\uparrow$)) Test |
|---|---|---|---|---|---|---|
| 1 | Joint Learning | 14.77 | 29.56 | 30.87 | 13.10 | 12.70 |
| 2 | Meta Learning | 14.74 | 28.58 | 29.92 | 13.89 | 13.67 |
| **This Work** | | | | | | |
| 3 | Task Aware Meta Learning (with implicit TCN) | 18.84 | **21.29** | **23.44** | **17.77** | **17.51** |

Table 6: Performance of models trained using different approaches on the ASR, MT and ST tasks using different datasets

| Task | Corpus | Train hours | Train Examples | Dev Examples | Test Examples |
|---|---|---|---|---|---|
| MT | WMT 14 | N/A | 4,592,289 | 3,000 | 3,003 |
| ASR | TED-LIUM 3 | 452 | 286,263 | 1,469 | 591 |
| ST | MuST-C | 400 | 229,703 | 1,423 | 2,641 |
| | Synthetic | N/A | 689, 103 | N/A | N/A |

Table 7: The data statistics of ASR, MT and ST tasks used in our experiments.

| Hyperparameter | *Single Modality* | *Multi Modality* |
|---|---|---|
| batching | dynamic | static |
| batch size | 2048 (tokens) | 16 (examples) |
| optimizer | adam | adam |
| adam_betas | (0.9,0.997) | (0.9,0.997) |
| lr_scheduler | inverse_sqrt | inverse_sqrt |
| lr | 2.0 | 2.0 |
| lr_warmup_steps | 16000 | 16000 |
| label_smoothing | 0.1 | 0.1 |
| dropout | 0.1 | 0.1 |
| lr_decay_rate | 1.0 | 1.0 |
| hidden_size | 512 | 512 |
| encoder_layers | 6 | 12 |
| decoder_embed_dim | 512 | 512 |
| decoder_layers | 6 | 12 |
| num_heads | 8 | 8 |
| filter_size(ffn layers) | 1024 | 1024 |

Table 8: Hyperparameter details for the experiments

