# OpenReview forum: "Learning without Forgetting: Task Aware Multitask Learning for Multi-Modality Tasks"
_ICLR.cc/2021/Conference — Reject_

### Official Review · AnonReviewer1 · 2020-10-24
**Multi-Task + FiLM + Transformers + Multi-Modal**

**Rating:** 4
**Confidence:** 5

**Review:**

This paper combines Feature-wise Linear Modulation (FiLM) with a single/multi-modal transformer for a joint multi-task neural network and applies the proposed model to two tasks, which are single-modal machine translation and multi-modal (speech/text) machine translation, speech translation, and speech recognition.

I have the following comments and reservations:
+ The paper is well-written and easy to follow.
- There is a plethora of closely related work that was not mentioned or discussed or compared to, e.g., Zhao et al.. 2018; Strezoski et al., 2019; Cheung et al., 2019, to name a few.
- The paper lacks comparison to strong multi-task baselines and compares the proposed model only to standard multi-task, transfer learning, and meta learning approaches and single-task state-of-the-art approaches that don’t have access to the additional data from the other tasks.
- The main title reads: “Learning without Forgetting”, yet there is no discussion or experiments with other methods that address the forgetting problem [Li & Hoiem, 2018].

As such, the work lacks sufficient novelty, is not well-placed within existing related prior work, and does not compare to adequate baselines.

References:

Brian Cheung, Alex Terekhov, Yubei Chen, Pulkit Agrawal, and Bruno Olshausen. Superposition of many models into one. arXiv preprint arXiv:1902.05522, 2019.

Zhizhong Li and Derek Hoiem. Learning without forgetting. IEEE Trans. Pattern Anal. Mach. Intell., 40(12):2935–2947, 2018. doi: 10.1109/TPAMI.2017.2773081.

Gjorgji Strezoski, Nanne van Noord, and Marcel Worring. Many task learning with task routing. arXiv preprint arXiv:1903.12117, 2019.

Xiangyun Zhao, Haoxiang Li, Xiaohui Shen, Xiaodan Liang, and Ying Wu. A modulation module for multi-task learning with applications in image retrieval. In Proceedings of the European Conference on Computer Vision (ECCV), pp. 401–416, 2018.

---

> ### Author Response · Authors · 2020-11-17
> **Comparison with previous work**
>
> Brian Cheung et. al. (2019) project the input to orthogonal sub-spaces based on the task information. Moreover, all the tasks share the same dataset varied by simple rotations which may not work for real world tasks where tasks might have substantial overlap. In Zhizhong Li et. al. (2018), the model is trained on various image classification tasks having the same input modality. They preserve the output of the model on the training example such that the parameters don’t deviate much from the original tasks. This is useful when the tasks share the same goal, e.g. classification. However, we train on a much more varied set of tasks, which might also have the same inputs with different end goals. Gjorgji et. al. (2019) propose to  apply a fixed mask based on the task identity. Our work can be seen as a generalization of this work. Xiangyun Zhao et. al. (2018) also relies on explicit task information. As compared to all these approaches, our model is capable of performing both task identification and the corresponding task learning simultaneously. It learns to control the interactions among various tasks based on the inter-task similarity without any supervision.

---

> ### Author Response · Authors · 2020-11-17
> **Strong Multitask baseline**
>
> After receiving the reviews, we extended Google's T5 model to the multi-modality settings by appending the task embedding to the input. This would serve as a strong baseline to test the effectiveness of the proposed modulation method. The extended T5 model fails in some cases where it completely switches the output from ST to ASR. For example, two consecutive checkpoints give a performance of 26.63 and 2.48 on the EnDe ST task, both being lower when compared to our model. Other than the performance, the bigger issue is that this model is unable to disambiguate the complicated tasks.

---

> > ### Author Response · Authors · 2020-11-23
> > **Final Results**
> >
> > The training for the model with joint space task embeddings has been completed. The final performance of the model is closer to the proposed model (-0.11 for ST and +0.08 for the ASR task). However, the stability issue still persisted, with the performance dropping considerably for some checkpoints. For e.g., the ST BLEU score is 28.31 for 510k steps, and 7.52 for 512k steps. The output is mixed between ASR and ST tasks. The proposed model did not suffer from this issue during any of our experiments. This shows that our proposed conditioning method is better than simply providing the task embedding as an input to the model.

---

### Official Review · AnonReviewer4 · 2020-10-28
**Room for improvement (good direction but poor execution)**

**Rating:** 4
**Confidence:** 4

**Review:**

This paper proposes a multi-task Transformer that does speech and machine translation within the same framework. It proposes task adaptive parameters by modulating model parameters to account for differences in tasks. The authors conduct experiments on speech and machine translation and show reasonable performance.

Honestly, this paper is lacking because this is just too incremental over the T5 model (which can train over 20 tasks without any task specific parameters). It feels quite lacklustre comparing to the many exciting multi-task or transfer learning works out there. The notion of training speech and language adds a little to the paper but I think it is insufficient to feel excited about.

The results in the paper don't look terribly good either. I'm also perplexed why the performance is even outperformed by the baseline. I also don't think it is really a good idea to reduce T5 to "OHV" (one hot vector) and completely remove the pretraining completely. Namely, "Unlike T5, we don’t initialize the models with the text embeddings from large pretrained language model (Devlin et al., 2018). Instead, we focus on establishing the importance of task information during MTL and having a single model for all the tasks." this is way too convenient and unconvincing. I think the authors should compare their model with T5 to have an apples to apples comparison.

I have also concerns about the practicality of modulating parameters and how this would cope over N task when N is large. The number of tasks in this paper is very small (compared to T5) and this is not well ablated in the paper. What about other adapter based approaches like projected attention layers (https://arxiv.org/abs/1902.02671) or hypergrid (https://arxiv.org/abs/2007.05891)? I think it's worth to compare the modulated parameters with other forms of multi-task adaptations of these transformer models.

Overall I recommend rejection. While this is a promising direction, this is poor execution. The proposed novelty of modulated parameters is also insufficient. Results are also not very exciting.

---

> ### Author Response · Authors · 2020-11-16
> **Comparison with T5**
>
> **Task Identification:** Unlike our implicit approach, T5 requires explicit information about the task which is not desirable in some settings. For instance, for multilingual speech recognition systems deployed in multi user settings such as in european parliament, it is better that the system performs both language identification and corresponding recognition simultaneously to facilitate smooth proceedings.
>
> **Removing Pre-Training & Results:**  In Table 2, we compare our work with T5 to show the importance of providing task information to the model. The “OHV” approach in Table 2 is exactly the same as the T5 approach without the pretraining step. We remove the pre-training for a fair comparison of implicit task information over explicit task information. The main results have been provided in Table 3 with multi-modality experiments where our proposed approach achieves SOTA results on the ST task.
>
> **Number of tasks:**  This is a great question. Since, we use multiple datasets for each task, the actual number of tasks is 10. We evaluated the performance and the learnt task embedding for each sub-task.The model is able to identify each sub-task(based on t-sne plots) based on the dataset and give decent performances on all of them. For example, **the single model achieves a WER of 7.5 on the librispeech tst-clean, 10.35 on MuSTC, and 11.65 on the TEDLIUM v3. For the ST task, the same model gives a BLEU score of 28.64 on the MuSTC test set, 27.61 on the IWSLT tst-2010, and 27.57 on the Europarl test set**. This shows that our proposed approach scales well with N. The new results will be added to the paper.

---

> ### Author Response · Authors · 2020-11-17
> **Performance compared to the baselines**
>
> We use the text-to-text(Table 1) experiments to show the importance of providing task information, we conducted our main experiments with the ASR, MT and ST tasks(Table 3). Our claim is that the approach suggested by T5 doesn’t extend to such settings. These tasks suffer from the confounding problem much severely since the input might be the same(ASR and ST), sometimes from the same dataset. After receiving the reviews, we extended the T5 model to the multi-modality settings by appending the task embedding to the input. The extended T5 model fails in some cases where it completely switches the output from ST to ASR. For example, two consecutive checkpoints give a performance of 26.63 and 2.48 on the EnDe ST task, both being lower when compared to our model. Other than the performance, the bigger issue is that this model  is unable to disambiguate the complicated tasks.

---

> > ### Author Response · Authors · 2020-11-23
> > **Final Results**
> >
> > The training for the model with joint space task embeddings has been completed. The final performance of the model is closer to the proposed model (-0.11 for ST and +0.08 for the ASR task). However, the stability issue still persisted, with the performance dropping considerably for some checkpoints. For e.g., the ST BLEU score is 28.31 for 510k steps, and 7.52 for 512k steps. The output is mixed between ASR and ST tasks. The proposed model did not suffer from this issue during any of our experiments. This shows that our proposed conditioning method is better than simply providing the task embedding as an input to the model.

---

### Official Review · AnonReviewer3 · 2020-10-28
**Automatic learning of task-specific representations without any explicit representation in an MTL setup**

**Rating:** 4
**Confidence:** 4

**Review:**

Summary:
This paper proposes a new framework that computes the task-specific representations to modulate the model parameters during the multi-task learning (MTL). This framework uses a single model with shared representations for learning multiple tasks together. Also, explicit task information may not be always available, in such cases the proposed framework is useful. The proposed framework is evaluated on various datasets spanning multiple modalities, where the MTL model even achieves state-of-the-art results on some datasets.

Pros:
1) The paper is well written and easy to follow.

2) The proposed framework achieves state-of-the-art results on some tasks/datasets.

Cons:
1) Seems like the implicit approach does perform more or less the same as the explicit approach. I believe the explicit approach is not something new and has been tried in various works. In that sense, the implicit approach is not exciting. I agree that the proposed framework would be able to learn task specific information without any explicit supervision, but this use case is not well explored in this paper. Would your approach scale to N (> 5-10) tasks?

2) From Table-2, OHV performs better than explicit TCN. The paper argues that providing OHV text labels is not always possible for tasks involving non-text modalities such as speech and images. But, I think it is possible to learn simple joint space task embeddings presentations for tasks from multiple modalities.

3)  From Table-3, the difference between explicit and implicit TCN methods is small. Statistical significance scores are not provided to know whether the improvements are real. I believe the explicit TCN approach is not something new so it would be better to know if implicit learning is strictly better than explicit.

Overall:
The paper is well written and easy to follow with SOTA results. However, I am concerned about the novelty aspect of the proposed framework. It is not clear from the current experiments whether the proposed task-specific representation learning is better than a simple OHV or simple multimodal joint-space task embeddings.

Questions:
1) Is implicit TCN better than explicit TCN in Table-3? Can you provide the statistical significance scores?

2) Instead of OHV, can you learn a join space task embedding representation? This paper: https://arxiv.org/pdf/1611.04558.pdf used special tokens for multilingual translation, can you think of extending to the multi-modal setup with speech and image? If not possible, why?

3)  Would your approach of learning task-specific representation learning scale to more task i.e., > 10 tasks?

Other comments:
1) It would have been easier to understand the implicit task information section if there is a corresponding pictorial presentation of the approach.

---

> ### Author Response · Authors · 2020-11-16
> **Will task-specific representation learning scale to more tasks?**
>
> This is a great question. The t-sne plots in the paper are drawn for only the three test datasets. However, we used multiple datasets for each of the ASR(Librispeech, Common voice, TEDLIUM, MuSTC-ASR), ST (MuSTC, IWSLT20, Europarl), and MT (WMT19, OpenSubtitles) tasks in the multi-modality experiments. We analyze whether or not our proposed approach is able to separate the data coming from these different distributions. As compared to data coming from different tasks, separating the data coming from the same task(generated from different distributions) is more difficult. Earlier, we observed task-based clustering(Figure 2), where the ASR, ST, and MT tasks were grouped into three main clusters. We found that within these clusters, the tasks have sub-clusters based on the source dataset. Hence, the model is able to identify each sub-task based on the dataset and give decent performances on all of them. For example, the single model achieves a **WER of 7.5 on the Librispeech tst-clean, 10.35 on MuSTC, and 11.65 on the TEDLIUM v3. For the ST task, the same model gives a BLEU score of 28.64 on the MuSTC test set, 27.61 on the IWSLT tst-2010, and 27.57 on the Europarl test set.** This shows that our proposed approach scales well with N. The new results will be added to the paper.

---

> ### Author Response · Authors · 2020-11-16
> **Is implicit TCN better than explicit TCN in Table-3?**
>
> Thank you for the question. Our claim does not hinge on Implicit TCN performing better than Explicit TCN. We wanted to show that even without explicit task supervision, Implicit TCN has similar performance as compared to Explicit TCN.

---

> ### Author Response · Authors · 2020-11-17
> **Learning joint space task embedding representation**
>
> Thank you for the question. The mentioned work is the same as one of our baselines in the text-to-text experiments. However, we conducted our main experiments with the ASR, MT, and ST tasks. Our claim is that the suggested approach doesn’t extend to such settings. These tasks suffer from the confounding problem much severely since the input might be the same(ASR and ST), sometimes from the same dataset. After receiving the reviews, we extended this work to the multi-modality settings by appending the task embedding to the input. The trained model fails in some cases where it completely switches the output from ST to ASR. For example, two consecutive checkpoints give a performance of 26.63 and 2.48 on the EnDe ST task, both being lower when compared to our model. Other than the performance, the bigger issue is that this model is unable to disambiguate the complicated tasks.

---

> > ### Author Response · Authors · 2020-11-23
> > **Final Results**
> >
> > The training for the model with joint space task embeddings has been completed. The final performance of the model is closer to the proposed model (-0.11 for ST and +0.08 for the ASR task). However, the stability issue still persisted, with the performance dropping considerably for some checkpoints. For e.g., the ST BLEU score is 28.31 for 510k steps, and 7.52 for 512k steps. The output is mixed between ASR and ST tasks. The proposed model did not suffer from this issue during any of our experiments. This shows that our proposed conditioning method is better than simply providing the task embedding as an input to the model.

---

### Official Review · AnonReviewer2 · 2020-10-28
**Proposed method is reasonable but more analyses would strengthen the paper**

**Rating:** 5
**Confidence:** 3

**Review:**

This paper proposes an way to incorporate task information into multi-task learning (MTL). The hope is that more explicit knowledge of task information will improve MTL, especially in cases where the model seems to confuse tasks (as shown in the example of speech translation in Table 1).

Pros:
- This is a timely contribution that addresses an important problem in MTL
- The proposed method is reasonable and relatively easy to implement: it basically involves adding a network that characterizes the task, and using its outputs to modify the MTL model's parameters.

Cons:
- While the results in Table 2 and Table 3 look promising, the presentation would benefit from more analyses experiments (see below). For example,  investigating whether the number of confused cases in Table 1 actually reduces, comparing whether the added parameters of the task-characterization network requires more data, exploring other ways to modulate the MTL model. Since the method is relatively straightforward, the paper would look stronger if it has more empirical analyses.

Additional questions and comments:
- Some claims need to be explained or clarified. For example in introduction the authors write: "However, providing task information is not straightforward for certain modalities such as speech and image. Also, providing explicit task information might not always be desirable, e.g., automatic multilingual speech recognition task." Why is task information not straightforward for speech? I understand it is harder to integrate the discrete label with the speech signal but it has been done before and the reason needs to be spelled out (i.e. don't assume the reader works in ASR.)
- Can you list out the same examples in Table 1 for the proposed systems? Did they solve the problem of confused tasks?
- Can you clarify how many new parameters are added for the different task-characterization networks, relative to the full network? It was also be interesting to fix to random weights in the task-characterization networks to see if the results are expected.
- Please discuss other methods for modulating the weights of the model, besides the one used. I think it's not necessarily to do experiments in those, but mentioning other potential techniques for modulation would make the idea more comprehensive, perhaps suggesting it as future work.

---

> ### Author Response · Authors · 2020-11-16
> **Experimenting with different Modulation methods.**
>
> As pointed out by Perez et al., 2018*,  the FiLM layer is a generalized conditional normalization layer which is robust to architectural changes. Thus, we we used FiLM layer as modulation layer.  As suggested, we will experiment with other modulation methods in the future work.
>
> *Ethan Perez, Florian Strub, Harm de Vries, Vincent Dumoulin, Aaron Courville.  FiLM: Visual Reasoning with a General Conditioning Layer, AAAI 2018.

---

> > ### Comment · AnonReviewer2 · 2020-11-20
> > **no need for experiment, just discussion**
> >
> > As I said in my original review, I'm not expecting additional experiments with different modulation methods, but simply a more detailed discussion of the different possible methods in the paper. This will help the reader have a broader understanding of the potential future work.

---

> ### Author Response · Authors · 2020-11-16
> **Number of parameters added**
>
> Thanks for the question. In the Explicit TCN which is a simple dense layer with the size same as the hidden size of the model(512), roughly 1500 new parameters are added. For the Implicit TCN has ~1 million additional parameters.However, simply increasing the number of parameters is not sufficient. For e.g., we trained several models by increasing the number of layers for encoder and decoder upto 16. However, these models gave inferior performance as compared to the reported models with 12 encoder and decoder layers.
> As suggested, we also initialized the weights of the TCN randomly and kept them constant during the training, the model failed to converge.

---

> > ### Comment · AnonReviewer2 · 2020-11-20
> > **thanks!**
> >
> > This is a nice additional result to include in the paper, thanks.

---

> ### Author Response · Authors · 2020-11-16
> **Solving the issue of confusion?**
>
> For each example mentioned in Table 1, the output of the proposed system matches the target closely and is not confused by other tasks. We will add the outputs in the final paper. We assumed that the improvement in performance would make it clear. For example, when ASR and ST tasks are trained with the same source language, the model  gives a very poor performance(<1 BLEU) for the ST task, while our proposed system gives a SOTA BLEU under similar settings.

---

> > ### Comment · AnonReviewer2 · 2020-11-20
> > **good!**
> >
> > Thanks for promising to add the examples. I think it will be instructive.

---

> ### Author Response · Authors · 2020-11-16
> **Providing task information with speech signal**
>
> Thank you for pointing this out. Adding task information to speech signals might be possible, however, it is not always desirable. For multilingual speech recognition systems deployed in multi user settings such as in european parliament, it is better that the system performs both language identification and corresponding recognition simultaneously to facilitate smooth  proceedings.

---

> > ### Comment · AnonReviewer2 · 2020-11-20
> > **thanks for the clarification, but please be even more specific if possible**
> >
> > Thanks for the clarification, but it would be helpful if you are more specific about the reasoning. For example, you said "it is better than the system performs...". What exactly do you mean by "better"? Do you mean there is better result empirically? To reiterate my comment in the original review: it is fine to make these claims but it would be helpful to explain the rationale behind them (i.e. don't assume the reader agrees or can infer your reasoning). Thanks!

---

> > > ### Author Response · Authors · 2020-11-23
> > > **'Implicit Task Information' approach performs task identification implicitly.**
> > >
> > > Thank you for pointing out the ambiguity. Extracting the task information from the input examples is 'better'/'desirable' for the settings which do not allow us to provide explicit task information during inference. For e.g., consider a speech-to-text translation system in a meeting with multilingual speakers, the 'Implicit Task Information' approach identifies the language of the input speech and produces the corresponding translation without requiring any user intervention. However, for the 'Explicit Task Information' approach, the user has to keep changing the source language setting depending on the speaker.

---

### Author Response · Authors · 2020-11-25
**Paper updated with new baselines, analysis, results & related work.**

We thank all the reviewers for providing valuable feedback. We believe that we have been able to modify our paper and improve the quality of our analysis due to the reviews. We have performed the following significant changes in the paper after receiving the reviews-

1. We have added a new baseline-
	- It helps us to comment on the efficiency of our proposed modulation network as a conditioning approach.
	- It helps us demonstrate why a task modulation based approach is required for a set of complex tasks.
2. We have shown that our approach scales well with the number of tasks.
3. We have provided new results for the other datasets used in our experiments, i.e., Librispeech, MuST-C, IWSLT, Commonvoice, and Europarl.
4. We have updated the ASR performance with the latest results, which beats the best results of the end-to-end deep ASR work mentioned earlier.
5. We add analysis about the number of additional parameters and their effect on the performance.
6. We have modified the writing style to express our motivation and findings lucidly based on the comments received.
7. We have updated the related works section to discuss and compare our work with relevant works based on the reviews.

We updated the paper with these modifications.

---

### Decision · Program_Chairs · 2021-01-07
**Final Decision**

**Decision:**

Reject

**Comment:**

This paper proposes an approach for improving MultiTaskLearning by providing a way of incorporating task specific information.

Pros:
1) All reviewers agreed that the paper is clearly written
2) Interesting to see a single model for AST, STS (speech-to-speech translation) and MT

Cons:
1) The work is not adequately compared with related work (some important references are also missing) - The authors did perform some additional experiments with T5  and pointed out some drawbacks but this needs to be explored a bit more.
2) The answers about scalability are not very convincing and need more empirical results.

Overall, none of the reviewers were very positive about the paper and felt that while this is a good first attempt, more work is needed to make it suitable for acceptance.